# Social Support Networks and Care for People Who Use Harmful Drugs

**DOI:** 10.3390/ijerph20043086

**Published:** 2023-02-10

**Authors:** Letícia Andriolli Bortolai, Ana Paula Serrata Malfitano

**Affiliations:** 1Occupational Therapist, Rio Claro 13501-900, Brazil; 2Post-Graduate Program in Occupational Therapy, Occupational Therapy Department, Federal University of São Carlos, São Carlos 13565-905, Brazil

**Keywords:** social support, social policy, professional practice, social occupational therapy

## Abstract

Background: The “problem of drugs” is a complex phenomenon with different social dimensions. Thus, the strategy to care for people who use drugs should consider their social support networks, which are defined here as dimensions that compose the social integration of people. Objective: In this paper, we investigate how social support networks are organized, structured, and constituted according to clients of a mental health service dedicated to treat alcohol and drug abuse. Methods: Participant observation was employed in a mental health service for three months, and six interviews and three groups of activities were conducted with local clients. Results: The results demonstrated that the social network of this group is composed of informal and formal social supports: the former includes family, religious institutions, and work, and the latter was represented by a few institutions. However, there are few supports that contribute to the social inclusion and participation of these clients. Conclusions: Care actions should expand social networks, helping to create more solid relationships, considering the macro and micro social-life dimensions. Occupational therapists can contribute to this process by driving their action toward social life, building more social participation strategies, and reconfiguring care and social meaning in everyday life.

## 1. Introduction

The use of harmful drugs has been understood as a complex phenomenon, involving multiple dimensions that make up the everyday lives of individuals [1]. Here, emphasis is placed on the aspects that compose a social perspective of drug use, going beyond understanding the specific effects of substances, and highlighting political and sociocultural contexts. The approach to drug use from a social point of view understands that there is not one substance or individual with similar elements, but rather different social and cultural contexts permeated by the use of different substances, and that without due attention to such differences, it is not possible to understand the theme in its entirety [2,3,4].

The theme present in the public debate has been observed with the concern that little has been achieved towards an effective resolution of what is understood by the “problem of drugs”, also called harmful use of drugs. This is a contemporary debate in which substance abuse demonstrates meanings that go beyond exclusively changes in sensations, in the degree of consciousness, or in the emotional state [5]. Two central points are presented: its growing incidence associated with challenges to find effective forms of “treatment”, when necessary, and its unilateral representation in the social imagination, classified as a “danger” to society, as something that needs to be curbed [6].

Considering the relevance and influence of political, historical, economic, and cultural contexts on drug use, different care possibilities are proposed. These are permeated by disagreements at different structural levels, demonstrating divergence regarding the possibilities to care for those who use harmful drugs [7]. There is no consensus on which direction the care for those who use drugs should follow—neither at a political level, nor at a professional level—regarding the approach, action methodologies, and techniques. However, it is essential to point out that actions aimed at coping “only” with the effect of drugs on individuals, regardless of the type of service, have been proven insufficient before the plots that intertwine the issue of drug use [8,9], although these actions are needed at times and in certain cases.

In this sense, it is crucial to recognize that drug use must be approached from a care shared among different actors, involving specialized social services of various sectors, extrapolating from a biomedical approach [10] to drugs. Therefore, it starts from a combined perspective between biological and social aspects to understand this complex and multifaceted phenomenon. Thus, to offer care that reaches the multiplicities related to drug use, it is believed that the concept of social support networks can be used as a key element to care for those people [11,12,13].

This approach starts from the notion that networks are understood as a set of relationships, formal and informal, in which individuals constitute the nodes. Informal networks are those characterized by the set of spontaneous interactions that arise in a given context, such as those composed of family, friends, and neighbors, among others [11]. Formal networks, on the other hand, can be considered as those that interconnect with social policies and services, with the participation of health units, schools and social assistance services, etc. [12]. Social networks are included in the dimension that comprises the social integration of individuals, with emphasis on work relationships and social networks, both primary and secondary [13]. They are structured by bonds between people, groups, and organizations built over time. 

When describing the so-called “social question”, Castel [13] analyzes the types of social insertions that are possible within the current system, considering work and social supports, and refers to the importance of social support networks, which can be considered elements that make up the social process. For Castel, the social network is shaped by the macro- and micro-structural dimensions. Macrostructure refers to the capitalist process and its exploitation, connected directly to work, which means social integration will be designed by the existence or not of work. On the other hand, microstructure is about personal support that can create different strategies to live. It includes formal support, such as health services, social care, schools, etc., and informal support, such as family, friends, church, collective organizations, etc. Based on this definition, the social integration to participate in this society is composed, in a dialectical way, by work and many different supports in each life.

Concerning the social network, Castel defines “zones” to characterize the social space where individuals can move about, having two main components that influence whether or not they belong to certain spaces: the axis of the work relationship (from stable employment to complete absence of work) and the axis of relational insertion (between enrollment in solid social networks and social isolation). In general terms, Castel discusses the organization of society and the existence of social fragility, which affect possibilities in the axes of work and social relations, which have led many people to a situation of vulnerability. People who use harmful drugs can be configured as one of the consequences of the situation of vulnerability, which has impacts on their personal and social network. Nevertheless, one cannot rule out the explanations that analyze the use of harmful drugs as a reflection of a fragile culture and society, permeated by economic interests, characterized, according to Castel, by a social isolation that weakens the possible supports for the social fabric of life.

Based on Castel’s theory, as well as on the combination of the mentioned concepts, professional actions are advocated through social support networks. This approach allows a more comprehensive understanding of the relationships that pervade the social environment where people are included. Emphasis is placed on social relationships and work possibilities since these are the main attributes that can offer conditions to (re)insert people into society, even if in a vulnerable and/or marginalized way. Many people who use harmful drugs are already socially disaffiliated, that is, they move between the “vulnerability zone”, defined by situations in which there is precarious or unstable work and/or fragility in the possibilities of social relationships, and the “disaffiliation zone”, which combines the absence of work associated with social isolation, and are inserted in a double rupture, resulting in social uprooting [13]. Castel also defines a fourth zone, the “assistance zone”, specifically regarding actions directed towards vulnerable and disaffiliated people, referring to formal (public services) and informal (charity) services. Therefore, using approaches that start from the restructuring of networks, or from the construction of new support possibilities, is a path that can be configured as an alternative or, at least, an amplifier to offer care actions.

Thinking of the complexity of drug use, the care offered must be multifaceted, and must include biological, social, economic, and cultural aspects, bringing together multiple strategies that can address needs more effectively [14]. To this end, social occupational therapy [15] is used as a methodological strategy of action to implement social care actions.

Social occupational therapy has been developing in Brazil since the 1970’s, based on its work in social settings. It is defined by a specific knowledge used in occupational therapy to deal with people lacking the social and economic resources to live. Social occupational therapy is practiced in a real context where people live and where political, economic, and cultural aspects of life shape and are shaped by everyday life. Social occupational therapy action is developed mainly along two axes: (a) as a theoretical and methodological framework for analysis and action with a focus on social life; (b) as the execution of specific social actions that are carried out beyond the field of health [15].

In this context, this study aimed to understand, from the perspective of drug users, how their social support networks are organized, structured, and constituted.

## 2. Methods

This study was carried out in a municipality with a population of approximately 200,000 inhabitants. Data were collected in a specialized public health service unit that monitors drug users.

This service unit performs, on average, 200 consultations a month directed towards people who use harmful drugs (licit and/or illicit). These people are assisted at this service unit in different ways: intensively—the person attends the institution for long periods during the day every weekday; semi-intensively—in the person attends the unit during specific periods of the week; and occasionally—for maintenance of care, when the person participates only in specific activities, such as groups and/or workshops. The service unit assists 40 people daily, on average. The team of this service is composed of 1 social worker, 4 psychologists, 1 psychiatrist, 1 occupational therapist, 2 nurses, 2 nursing technicians, 1 psychology intern, 1 administrative technician, 1 receptionist, and 1 general services assistant. The services offered are carried out in different stages, namely, therapeutic groups, home visits, workshops, external activities, coexistence activities, individual assistance, and medical consultations.

As a strategy to conduct the study fieldwork, methods that would provide an in-depth description and interpretation of its contents were used with a view to a comprehensive approach to reality [16]. The data were collected through triangulation between 3 different sources.

First, there was observation of the institutional dynamics by remaining in the institution for 3 months and registering the activities in a field diary. The observation was conducted by the first author of this article, related to all institutional dynamics. Following this, joint activities were carried out with the clients through thematic workshops, seeking resources beyond the verbal report. Finally, individual interviews were conducted with 6 service clients.

The proposal to carry out activities is something that the participants were already used to, because within the institution, many approaches are made using this resource. With the immersion in the field and knowledge about the context of the clients in the institution, considering the importance of their active participation in the research process, the following activities were proposed:

Activity Workshop 1—importance clothesline: each participant was asked to respond to the following triggering question: “Who or what helps me in times of difficulty?” Graphic materials were made available, and a response was requested, which could be provided through images and/or words. The participants were given the possibility to express themselves through drawings, symbols, collages, words, sentences, etc. They were allowed 15 min to carry out the activity, and afterwards, each one gave a presentation of what was represented in their activity and placed the product on a clothesline. With that, we sought an individual moment of reflection and, later, a collective discussion with the joint construction of visions. At the end of the presentations, the similarities and differences between the opinions expressed were discussed. Five participants, all male, aged 20–50 years, attended this activity.

Activity Workshop 2—city resources: Each participant received a printed map of the city containing a simplified division of the neighborhoods, and, individually, for 20 min, indicated the places where they circulate and which environments they consider important in their everyday lives. They could use drawings, symbols, etc., to answer the following triggering questions: “Which city locations are important to me? Where do I go? To do what? Where can I get help?” They were instructed that they could draw, write, or make collages. Next, each participant presented their product, and a group discussion was held on the differences between the places; who knows the places mentioned, how each one makes use of the city, and their references for times of difficulties were discussed. Eleven participants, both male and female, aged 20–65 years attended this activity.

Activity Workshop 3—the mental health service: a collective preparation of 2 graphic materials that contemplate the following questions was proposed: “How do I see this service unit? What would a location that could help me with my needs look like?” The participants could use any material they wanted to compose a graphic representation, and a drawing of 2 houses was given to them. One of these drawings represented their view of the service unit and, later, they created an ideal place that would meet their needs. After 40 min of preparation, the presentation of the built places was requested for group discussion. Eleven participants, both men and women, aged 20–65 years, attended this activity.

A total of 20 people participated in the activities, which means 7 people participated in more than 1 activity.

To facilitate the analysis, the meetings were audio recorded, with authorization of the participants, totaling 3 h and 34 min of recording, in 3 different meetings.

Subsequently, individual interviews were conducted with the aim of identifying and analyzing the points that characterized the social support network of the participants, including verifying whether the services were present in this list and how these individuals relate to drugs within their everyday lives. The invitation was made to all people present at the service unit under intensive follow-up, and the objective of the study was explained. A total of 6 participants contributed to this stage: 5 men and 1 woman, aged 27–63 years, in different stages of treatment; some of them had already attended the service at other times and others were there for the first time. Each interview lasted from 45 min to 1 h and 15 min. The interviews began with the following triggering question: “What and/or who do you rely on when you need help regarding drug abuse?”

Results from observations and field diary records, associated with data from the interviews and activities carried out, were analyzed. Based on these sources, a thematic analysis [17] was performed by ordering and classifying the data, which are built from questions based on the objective of the work, with the chosen theoretical fundamentals as a reference. Both authors were involved in the thematic analyses, reading the transcriptions, proposing themes, and discussing them. The whole analytical process was informed by the comprehensive approach to reality [16], focusing on giving one interpretation, among others, about that reality. The analysis was also informed by social occupational therapy [15], specifically regarding the social issues involved. This analysis, via a socio-historical approach, seeks to apprehend the local reality and refuses to understand social problems as merely health issues. The authors highlighted the social aspects of the results found.

## 3. Results

In view of the triangulation of field information, the data came from three collections of resources. It is worth noting that the most explicit information was collected through the interviews. The activities were very important to obtain a more collective opinion discussed by the participants, to be either confirmed or not confirmed by them. The institutional observation allowed us to understand the components mentioned by the participants more comprehensively in both activities and interviews. It would be helpful to have conducted a deeper observation using alternative methodological strategies, such as institutional ethnography, to obtain more details about the conditions of the people who attend this type of service and their requirements. In our field, the observation gave a general idea about the dynamic of their relationships through the interviews and activities, in this order, as the sources that provided more information.

The following components of the social support networks of users of harmful drugs who attend a mental health service were identified: family, religious institutions, work, and mental health institutions. This demonstrates the presence of formal and informal elements in the composition of their networks.

### 3.1. Family

Regarding informal relationships, a common factor among the participants was the importance of their families. In the six interviews carried out, all participants named one or more family members as an important element in their support network. Some of them mentioned people in their nuclear family, such as mothers, fathers, or siblings, while others named their partners/spouses. These people were also mentioned during the group activities.

Family members were presented ambiguously, sometimes strengthening the support network, sometimes weakening it, and some participants pointed to the family as a hindrance in their “recovery”/“rehabilitation” process.

*People close to a drug user should, in the first place, support those who live with them. And if the people who are in direct contact with the drug user don’t have minimum knowledge and understanding about the subject, they won’t support them […] they will not be able to restructure […] I believe the first thing would be family support*.(Participant 4, Interview)

The debate regarding the ambiguity of the role played by family in an informal support network has also been observed in previous studies that focused on the theme of drugs. The family can be considered as a protective factor against drug abuse and situations of social vulnerability. In contrast, it can also be characterized as a risk factor for triggering stressful events [18,19,20]. Some family ties can be broken as a result of drug use, but these are permeated by ambiguities and conflicts. According to different studies, one of the justifications for the instability of relationships is related to the fact that families do not believe in the recovery process and are worn out by the recurrent relapses during treatment. Biegel, Katz-Saltzman, and Townsend [21] investigated the relationships between family members and drug users and reported that the quality of support influences treatment outcomes and the involvement of people in the provision of care.

The emphasis given to the family underscores the importance of thinking about care that includes the affective relationships that are relevant to those individuals, so that they can have the opportunity to provide effective support for people who use harmful drugs.

In addition to the nuclear family, other informal relationships were mentioned as components of social networks. The so-called “bad friendships” by some participants are an example of this. In a study conducted by Kantorski and Mielke [20], friends are clearly pointed out and highlighted as playing a central role in the possibilities of support for drug users. Added to this discussion is the influence of the social contexts in which the individuals are inserted, which can be characterized by territories marked by situations of vulnerability and little access to social policies.

*What can make things difficult are the friendships! Friendships also make it difficult for people to quit, because no one is forced to do anything, do you understand? […] I’m not going to use it, but I’ll be together, and then one time you end up using […] That’s why there are many people who come here and say, “Ah, there’s no way man, I walk out the door and the drug selling point is there”. Wherever I go there’s a drug dealer. They all know me, they’ll call me, we’ll talk, other people are using it, so nobody is forced to use it, but one time … So, it’s a cycle, you look for it, so I think that’s what makes it a little difficult, too. It’s difficult, the person refuses today, refuses tomorrow, but eventually they’ll take it*.(Participant 1, Interview)

Family, in this broader sense, exhibits the centrality of informal networks configuring the everyday lives of people. This finding composes the social network, defined by Castel [13], of the participants of this study.

### 3.2. Religious Institutions

Another social support mentioned by the participants was religious institutions, in several segments, especially participating in groups associated with churches. During the interviews, one participant emphasized that attending church does not necessarily refer to the issue of religiosity or specifically to spiritual support, but rather to the acceptance they find in that environment, which makes them feel accepted, and they are not being judged.

*I go to church, but I have my doubts about the existence of God. I’m drug crazy … But at least in church there are no drugs, there are no people wanting to talk, judging me … Like it or not, the church is a place where people welcome me … Bring me a word of comfort*.(Participant 4, Interview)

This report corroborates the results of other studies that directly discussed this issue [4,20], which emphasize that religion is often a support alternative for the gaps present in the everyday lives of these individuals. Religious institutions, or more specifically the people who compose them, represent possibilities of non-stigma and belonging, and create a meaning for being there, just as, many times, drug use represents participation [2]. However, religious institutions were also mentioned by some participants as places where they did not receive support and were excluded because of their history of drug use.

The contradictions found both in the discussion related to the family and in the fact that the church is, or not, a place of support indicate how much the theme of drugs is composed of different elements depending on the people who make up such institutions, positively or negatively influencing each person’s support networks. Souza, Kantorski, and Mielke [19] also stated that individuals may have difficulties in classifying certain bonds, considering that the same relationships can also be sources of stress or assume different characteristics depending on the situation they experience.

Thus, during care, it is necessary to reflect on subjectivities, emphasizing what a particular individual considers important in their life context. Hence, possibilities of generalized approaches are difficult, and singular attention is relevant. Along the same lines, the findings of Machado, Modena, and Luz [18] corroborate this perspective by demonstrating that drug users who seek mental health services have different needs, demands, and expectations regarding the care offered, which can be associated with their individual experiences of drug abuse and with the social and structural processes they experience. Therefore, this finding highlights the relevance of the informal social network, composing the possibility of support to be integrated into society, as defined by Castel [13].

### 3.3. Work

Another theme brought up by the participants during the interviews was work, also considered as an important support possibility and, many times, placed as the main organizing element of everyday life. Psychosocial rehabilitation [22] has historically discussed work in association with citizenship and emphasizes the work axis as a relevant part of care in mental health services. Services must act to produce new forms of social inclusion for people, and, in the current society, work is characterized as an essential element for social participation. The emphasis given to work is combined with Castel’s theory [13] on the very definition of a support network. According to Castel, work is the central element in the social support network of individuals, considering the constitutions of capitalist society and, consequently, its centrality. It is through work, associated with personal and social relationships, that social integration occurs. Thus, the participants’ reports show how work organizes life and is central to the fabric of other supports and relationships.

*Work is important, very important […] when I wasn’t working, my head was empty […] I would drink till I slept and would wake up to drink again. I used to wake up, hang about, heard the gate opening and would go out … So, without working, you’ve got nothing to do, stay home, you can’t. So, I think working is essential*.(Participant 1, Interview)

Work was mentioned as the main element of social belonging, but with great difficulty in access and permanence. Working demonstrates the connection between macrostructures and social precariousness [15], and also with the personal support network [13], as it is related to ‘doing’ and ‘meaning’ in our society. Work is part of the macro- and micro-spheres, similar to individual trajectories, composed mostly of social relationships marked by stigmatization, disqualification, and marginalization [19].

### 3.4. Mental Health Institutions

Regarding the services that make up social support networks, that is, the formal networks, in addition to work, different mental health institutions accessed by the participants were mentioned.

All interviewees mentioned the outpatient service they attend daily as a fundamental support for their life possibilities (the field of this research). For some of them, it was the only support available within their current relationships at that moment.

*I go to bed thinking that tomorrow I’ll come here, that I’ll see so-and-so. It’s a motivation to live, to think about my day, that I have commitment, responsibility, that I have to be here, it’s my obligation, that they are waiting for me … It is a life motivation. For me, it’s been good, it’s having an effect, both here and at home […] I leave here with enthusiasm […] so these are things that make sense to us, which we don’t have at home*.(Participant 5, Interview)

However, the centrality of certain institutions shows the precariousness of access for those people to other social services. There were no reports of places for leisure, culture, education, socializing, etc., that could compose their circulation and everyday life, which would certainly have an impact on their support networks.

*I come here, I’ve made friends with everyone, I talk to everyone, so … The people are nice, receptive, just arriving here is already nice […] if I stay all day, I can have lunch*.(Participant 1, Interview)

This fact was confirmed during the activity workshops, especially in the second one where the city map was explored. We observed that the circulation of these individuals in the city is very restricted, highlighting their social precariousness and little access to the right to urban mobility [23]. The reports and drawings they produced indicated that the wider the social networks, the greater the circulation of these people. The people who showed greater circulation through the territories are those who had a slightly stronger support network. This is exemplified in Figure 1 and Figure 2, where the participants described on a map (Activity Workshop 2) the places they know in the city.

*I go to Gaviões to play soccer. And when I need help, I talk to my ‘protector’*.(Workshop 2, transcript of the group discussion. Figure 1)

Within the scarcity of social policies, only the presence of specialized health institutions is observed, overloading the social function of these services as well as restricting the possibilities of access and experiences for people. As one of the consequences, the mental health service unit becomes the place to eat, socialize, and be treated, occupying a space that escapes the logic of community and territorial care, which reveals the precariousness of other policies for this socially vulnerable population.

In addition, the “inpatient/recovery clinics”, as they are called by the participants, comprise another theme that appeared recurrently in the participants’ speeches, probably because they have already been admitted to these institutions. In these clinics, the most common treatment proposal is abstinence, considered as a standard treatment efficacy by these institutions; however, this has not been achieved by any of the hospitalized participants, causing great frustration. There were reports of suffering due to deprivation of liberty.

*You’re withdrawn from society, for example, for 3 or 6 months, you’re withdrawn and you’re not going to use drugs at all … There’s no way, you’re locked there. But, at the same time, it causes you such a great revolt, a very great revolt … You lose control over your life, it doesn’t encourage you to control your life, your desires … You have to obey the rules. You’re deprived of your family, of taking care of your financial life … You are deprived of everything […] And you leave there feeling disgusted, because several months of your life have been taken and your life is all behind schedule, your problems are all there, they have multiplied, you leave in desperation to try to get your life right and put it in order to start walking again […] So there are a lot of people there with 10, 20 hospitalizations … So that means it doesn’t solve the problem […] I met people at the clinic who have been there for two years, absurd. And when they leave, they are going to use drugs again, for sure […] And inside, there’s humiliation, there’s violence*.(Participant 4, Interview)

## 4. Discussion

Bearing in mind the dimensions listed as significant by people undergoing treatment for drug abuse, care actions in occupational therapy and the relevance of focusing on social support networks to build more social participation strategies are discussed.

The complexity of the “drug problem” [1] requires multiple approaches that are not limited to the biomedical aspects of action, especially when dealing with socially vulnerable populations. Considering that care in occupational therapy may involve biomedical, person-centered, tacit, and collective dimensions [14], occupational therapists have tools to direct their actions to the expansion of the individuals’ social networks, thus helping to create more solid relationships, considering the macro and micro social-life dimensions.

The participants showed important points in their networks, presenting possibilities of actions in occupational therapy. All aspects related to the support network are connected to ‘doing’ and ‘meaning” in everyday life, presenting possibilities for professional occupational therapeutic work to increase their social support network. This might be approached through these elements presented by the participants, such as proposing new possibilities and creating opportunities of access that could result in a larger network. In other words, we advocate that occupational therapists act specifically on the social networks, as proposed by social occupational therapy in its methodological professional resources [24].

Studies in this field have shown the influence of family and friends [20,21], religious institutions [4,18,20], and work [19,22] on the everyday life of people in mental health services. Even with other groups, such as refugees [25], the centrality of social networks is demonstrated. Occupational therapists are recognized as one of the professionals who have the ability to work through and with social networks [26].

We have used social occupational therapy, which emphasizes the need to design care actions centered on the social dimensions of life [15]. Its proposals suggest combining resources in the social field and dynamizing the support network to advocate for the expansion of care actions in occupational therapy, aiming at the social participation of individuals. “The articulation of resources in the social field comprises an array of actions carried out at the individual level, moving up to the group and collective levels, and eventually to the levels of politics and management. The strategy consists in managing the practices at different levels of assistance involving common goals and using the possible resources, understood as the financial, material, relational and affective devices, either macro- or micro-social, to compose the interventions” [24] (p. 174). The dynamization of the support network “aims to map, disseminate, and consolidate all programs, projects, and actions aimed at this population group and/or its community to foster interaction and integration, articulate the different sectors and levels of intervention, and facilitate the effectiveness and direction of the strategies” [24] (p. 174). Based on these strategies, the possibility of action with a focus on the meaning of each person’s networks is considered, seeking to transform experiences by strengthening support and creating possibilities for greater social participation [2].

## 5. Conclusions

Limitations to this study include the fact that it was conducted at only one unit service with a small number of participants. It could be benefited by a deeper institutional observation process. Furthermore, future research with deeper discussions on people’s social networks, care, and contemporary society could add more elements for this topic. Other places, strategies of collecting data, including activities and other visual methodologies or ethnographies might be discussed as possibilities of improving this contemporary and challenging theme. However, based on critical, participatory, and qualitative work, this study could be understood as research that offers elements for continued dialogue to build a comprehensive perspective of this reality.

The social networks of people who use harmful drugs are composed of informal elements, such as family and religious institutions, as well as of formal elements, such as work and mental health institutions. However, they are scarce and often permeated by conflicts. While some may offer support, there are also ambiguities and tensions in the relationships. This is found in both the informal context, in a more local way as in the case of families, and the formal context, constituted by services, where conflicts are present in different ways, whether in models, directions, and possibilities for monitoring, or in difficulty of access.

The reality of the participants of this study clearly showed the precariousness of social policies, evidenced mainly by the absence of reports on spaces for leisure, culture, coexistence, and education, among others, which could certainly enrich their everyday lives, as well as by their circulation restriction in the urban space. Added to this precariousness, the social stigma they experience for being “drug users” keeps them away from the few possibilities they could have. Thus, a double precariousness is observed, which marks their everyday lives and may hinder the treatment they seek in institutions. It demonstrates the necessity to discuss social policies targeted towards this population that are not based on healthcare, but come from an intersectoral perspective, addressing social rights for all. Specifically, for the healthcare units, the data show the necessity to include the social network as part of professional actions, emphasizing the everyday life of people in their social context.

Therefore, care for this population cannot be centered exclusively on the reproduction of biomedical models, or on the best therapy to curb addiction. It needs to include elements aimed at strengthening the social support networks of these individuals. Thus, the proposition of a professional action through the expansion of social networks can favor the composition of different types of support, so that it can effectively help to create a more solid network, encompassing needs that go beyond health care.

Occupational therapists have the skills to perform such actions if they direct their focus to social life. Social occupational therapy can be a theoretical–methodological framework for this achievement, which highlights a social view of society, proposing action to create and/or increase the social support network.

## Figures and Tables

**Figure 1 ijerph-20-03086-f001:**
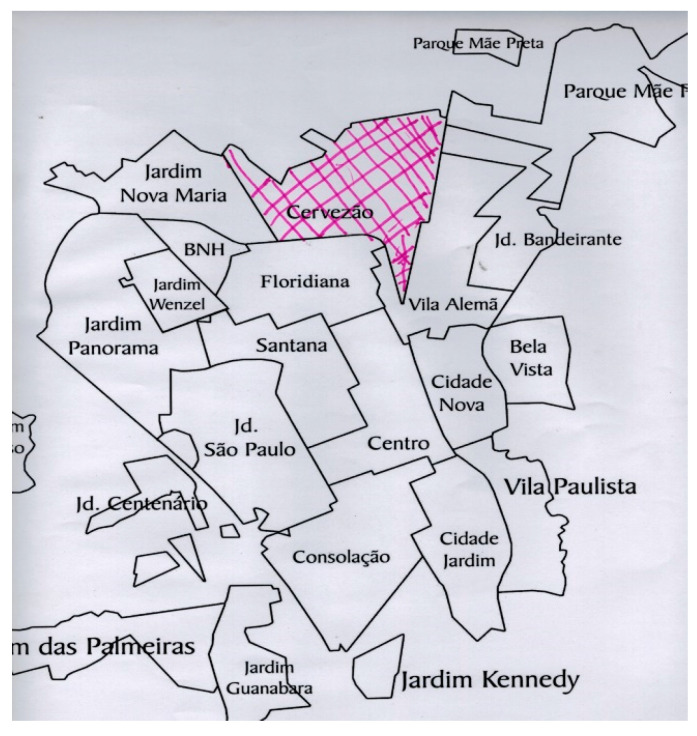
Product of “Workshop 2—Importance clothesline”. The colored part is the place where the person circulates in the city.

**Figure 2 ijerph-20-03086-f002:**
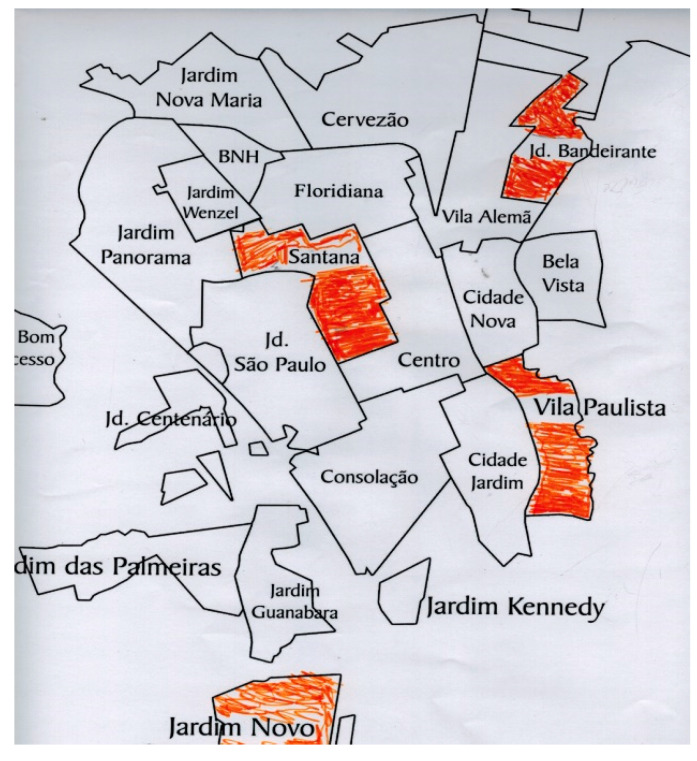
Idem Figure 1.

## Data Availability

This study is based on the Master thesis of the first author, supervised by the second author, in the Post-graduation Program in Occupation Therapy, Federal University of São Carlos, Brazil. The integrality of the master thesis is available on: https://repositorio.ufscar.br/handle/ufscar/12471.

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
