# Peer review of "Social Support Networks and Care for People Who Use Harmful Drugs"

_ijerph, 2023, doi:10.3390/ijerph20043086_

Round 1

Reviewer 1 Report

Does the introduction provide sufficient background and include all relevant references?

The authors theoretically ground their definition of the ‘social,’ considering broad factors.

In the introduction, I found it difficult to discern what information/perspectives were attributed to another source (other than direct citations) and what was a novel interpretation from the authors. Description of Social Occupational Therapy would be helpful.

Copyediting would strengthen this section.

Are all the cited references relevant to the research?

This is suitable.

Is the research design appropriate?

This is an interesting and innovative approach. It would be great to see a second paper describing the development of this methodology.

It would be helpful to either cite sources that provide a rationale for the design or explain the rationale for this novel approach.

Are the methods adequately described?

It is stated “This service unit performs, on average, 200 consultations a month directed to drug (licit and/or illicit) users” – how many ‘drug users’ would be served (it’s not clear to me if each consultation is for a different person, or if a single person would receive more than one consultation).

I recommend describing the observational aspect of data collection in more depth. What and who were being observed?

How many participants in the activities in total? Were there participants who attended more than one activity?

Did anyone participate in both interviews and activities?  

What approach was used for analysing the three sources of data collection in relation to one another? What was the process for undertaking the thematic analysis? What steps were taken and who was involved? How specifically did Social Occupational Therapy influence the analysis?

Are the results clearly presented?

The sections are clearly presented and the select quotes are rich in their descriptions.

It would be helpful to know how the three data sources contribute to the findings in a little more depth. Was there consistency in the type of information shared across all methods? Contradictions? How did the observation period inform the design of the activities or interpretation of findings?

The analyses of ‘religious institutions’ and ‘work’ are scant in specific relation to the data. Instead, secondary sources are cited, which might be better suited for the discussion section.

The quote about work did not convey, to me, an aspect of the social; rather it seemed to be about the ‘doing.’

In section 3.4, there seems to be opportunity for the intersection of the setting (place) and social (people in those places) – perhaps helpful to acknowledge that participants sometimes discuss place and people as intertwined? Otherwise, the analysis seems to diverge from the ‘relationships’ towards ‘institutions.’

Figures 1 and 2 need to be described.

On p. 2, it is stated that “[Social networks] are structured by bonds 67 between people, groups, and organizations built over time. The quote on p. 8 appears to focus on a clinic. Building my earlier comment recommending delineation of relationships and institutions, I think the paper (and the analysis specifically) would benefit from increased focus on the ‘network.’ The analysis appears fractures, with some aspects of the social being discussion within each themes and other aspects of the social are absent.

Are the conclusions supported by the results?

The Discussion introduces occupational therapy, but without clear links to the findings.

In the conclusion, I would have liked to see some discussion about how the data collected across sources intersected.

I also would have liked to see a section discussing the limitations.

Given my recommendations for some added content, the authors might consider a more succinct description of the ‘social’ at the beginning, which would provide space to provide more details about the methods. As mentioned, I recommend the researchers revisit the findings to more fully develop the findings that emerge from the data, rather than how aspects of the data relate to the broader literature base.

I enjoyed reading this paper and see it offering a novel contribution to the knowledge base.  

Author Response

Dear reviewers, we really appreciate  evaluation of our article “Social support networks and care for people who make use of harmful drugs”. We highlight that the referees’ notes have enabled us to improve our work.

The manuscript has been edited for proper English language, grammar, punctuation, spelling, and overall style by an accredited editor.

We have adjusted the manuscript following your guidelines. For further explanation, we would like to highlight some points:

Reviewer 1:

Description of Social Occupational Therapy would be helpful.

Answer: We have included a definition of Social Occupational Therapy in the Introduction section.

This is an interesting and innovative approach. It would be great to see a second paper describing the development of this methodology.

It would be helpful to either cite sources that provide a rationale for the design or explain the rationale for this novel approach.

Answer: We thank the reviewer for the comments.

It is stated “This service unit performs, on average, 200 consultations a month directed to drug (licit and/or illicit) users” – how many ‘drug users’ would be served (it’s not clear to me if each consultation is for a different person, or if a single person would receive more than one consultation).

Answer: Thanks for the comments. We have added a paragraph explaining that people can attend the service unit in intensively, semi-intensively or occasionally, and that unit assists 40 people daily on average.

I recommend describing the observational aspect of data collection in more depth. What and who were being observed?

Answer: We have added text informing that the observation was conducted by the first author of the article regarding all institutional dynamic. 

How many participants in the activities in total? Were there participants who attended more than one activity? Did anyone participate in both interviews and activities?  

Answer: To clarify this point, we have added the following: “A total of 20 people participated in the activities, which means that seven people participated in more than one activity.

What approach was used for analysing the three sources of data collection in relation to one another? What was the process for undertaking the thematic analysis? What steps were taken and who was involved? How specifically did Social Occupational Therapy influence the analysis?

Answer: We have added text informing that both authors were involved in the analytical process, following the theoretical principles of the comprehensive approach to reality, based on Bourdieu. The influence of Social Occupational Therapy on the analyses was also explained.

Results: The sections are clearly presented and the select quotes are rich in their descriptions.

It would be helpful to know how the three data sources contribute to the findings in a little more depth. Was there consistency in the type of information shared across all methods? Contradictions? How did the observation period inform the design of the activities or interpretation of findings?

Answer: Thanks for this comment, we really appreciated it. In brief, we have added a paragraph addressing this aspect.

The analyses of ‘religious institutions’ and ‘work’ are scant in specific relation to the data. Instead, secondary sources are cited, which might be better suited for the discussion section.

Answer: We have chosen to associate some findings with discussions in this topic, highlighting the discussion about the professional occupational-therapeutic practice with this population.

The quote about work did not convey, to me, an aspect of the social; rather it seemed to be about the ‘doing.’

Answer: Thanks for highlighting this point. We associate the work dimension of life with doping in the everyday life.

In section 3.4, there seems to be opportunity for the intersection of the setting (place) and social (people in those places) – perhaps helpful to acknowledge that participants sometimes discuss place and people as intertwined? Otherwise, the analysis seems to diverge from the ‘relationships’ towards ‘institutions.’

Figures 1 and 2 need to be described.

Answer: A brief description has been added.

On p. 2, it is stated that “[Social networks] are structured by bonds 67 between people, groups, and organizations built over time. The quote on p. 8 appears to focus on a clinic. Building my earlier comment recommending delineation of relationships and institutions, I think the paper (and the analysis specifically) would benefit from increased focus on the ‘network.’ The analysis appears fractures, with some aspects of the social being discussion within each themes and other aspects of the social are absent.

Answer: The idea of occupational therapy work based on the creation and/or increase of a social support network was reinforced.

I also would have liked to see a section discussing the limitations.

Answer: Thanks for pointing out that. It has been included in the Conclusion section.

I enjoyed reading this paper and see it offering a novel contribution to the knowledge base. 

Answer: Thank you so much! We really appreciated your contribution.

Reviewer 2 Report

This is an interesting study in which the authors discuss the role of social support on drug use, and I think it could be considered for publication, after addressing the following questions:

1. I do not understand the relationship of Castel's theory to explain the influence of social networks on individual drug use, because Castel's theory only explains social phenomena and cannot be equated with social networks. Therefore, I think it is necessary to add a paragraph after line 97 to describe the relationship between Castel's theory and social networks.

2. I think it is necessary to add some literature on the relationship between social networks and drug use behavior after line 58.

3. You emphasize the importance of work in Castel's theory, but neglect to explain how family and Religious institution figure in Castel's theory. I think it is necessary to emphasize.

4. line 282 mentions that the formal network, can I understand that work, Religious institution, and family are the informal network. how is this differentiated in Castel's theory?

5. I think the diagrams in lines 308 and 310 need to be redrawn, and the authors can rely on mapping software such as Arcgis.

6. The author should add a paragraph after line 346 to discuss the conclusions drawn compared to previous studies.

7. the authors should add a paragraph in line 363 that states their expectations for future research.

8. similarly, the authors should add policy recommendations.

Author Response

Dear reviewers, we really appreciated your evaluation of our article “Social support networks and care for people who make use of harmful drugs”. We highlight that the referees’ notes have enabled us to improve our work.

The manuscript has been edited for proper English language, grammar, punctuation, spelling, and overall style by an accredited editor.

We have adjusted the manuscript following your guidelines. For further explanation, we would like to highlight some points:

Reviewer 2:

  1. I do not understand the relationship of Castel's theory to explain the influence of social networks on individual drug use, because Castel's theory only explains social phenomena and cannot be equated with social networks. Therefore, I think it is necessary to add a paragraph after line 97 to describe the relationship between Castel's theory and social networks.

Answer: Castel’s theory delimits both dimensions of social integration: work and personal supports. For Castel, the social network is shaped by the macro- and micro-structural dimensions. Macrostructure refers to the capitalist process and its exploitation, connected directly with work, which means social integration will be designed by the existence or not of work. On the other hand, microstructure is about all personal support that can create different strategies to live. It includes formal support, such as health services, social care, schools, etc.; and informal support, such as family, friends, church, collective organizations, etc. Based on this definition, the social integration to participate in this society is composed, in a dialectical way, by work and many different supports in each life. Considering this concept, we understand the experience of the use of harmful drugs as an element that fragilizes the social support network, considering the social stigma and the ruptures with individual supports.

To clarify this point, we have added more details about Castel’s definition of social support network. 

  1. I think it is necessary to add some literature on the relationship between social networks and drug use behavior after line 58.

Answer: It has been added the references already present in the text.

  1. You emphasize the importance of work in Castel's theory, but neglect to explain how family and Religious institution figure in Castel's theory. I think it is necessary to emphasize.

Answer: Considering Castel’s definition of social network, as an element to define social integration (and vulnerability and disaffiliation), family and religious institutions are considered as part of the informal network, constituting these phenomena. Two sentences (one at the end of subitem Family and one in subitem Religious Institutions) have been added to highlight this proposition.  

  1. line 282 mentions that the formal network, can I understand that work, Religious institution, and family are the informal network. how is this differentiated in Castel's theory?

Answer: It is in accordance with Castel’s theory.

  1. I think the diagrams in lines 308 and 310 need to be redrawn, and the authors can rely on mapping software such as Arcgis.

Answer: The figures show photos produced from the Activity Workshop 2, made by the research participants.

  1. The author should add a paragraph after line 346 to discuss the conclusions drawn compared to previous studies.

Answer: A paragraph has been added to this end.

  1. the authors should add a paragraph in line 363 that states their expectations for future research.

Answer: We thank the reviewer for this comment. It has been included in the paragraph about the limitations to this study and future research.

  1. similarly, the authors should add policy recommendations.

Answer: Thanks for this comment. Some suggestions have been included in the Conclusion section, specially highlighting the necessity to invest beyond healthcare.

Round 2

Reviewer 2 Report

The author did adequate reviews, i think the manuscript could be accepted.